# Phonon-enhanced photothermoelectric effect in SrTiO$_3$ ultra-broadband photodetector

Xiaowei Lu[1,2], Peng Jiang[1] & Xinhe Bao[1]

The self-powered and ultra-broadband photodetectors based on photothermoelectric (PTE) effect are promising for diverse applications such as sensing, environmental monitoring, night vision and astronomy. The sensitivity of PTE photodetectors is determined by the Seebeck coefficient and the rising temperature under illumination. Previous PTE photodetectors mostly rely on traditional thermoelectric materials with Seebeck coefficients in the range of 100 $\mu$V K$^{-1}$, and array structures with multiple units are usually employed to enhance the photodetection performance. Herein, we demonstrate a reduced SrTiO$_3$ (r-STO) based PTE photodetector with sensitivity up to 1.2 V W$^{-1}$ and broadband spectral response from 325 nm to 10.67 $\mu$m. The high performance of r-STO PTE photodetector is attributed to its intrinsic high Seebeck coefficient and phonon-enhanced photoresponse in the long wavelength infrared region. Our results open up a new avenue towards searching for novel PTE materials beyond traditional thermoelectric materials for low-cost and high-performance photodetector at room temperature.

[1] State Key Laboratory of Catalysis, CAS Center for Excellence in Nanoscience, Dalian Institute of Chemical Physics, Chinese Academy of Sciences, 116023 Dalian, China. [2] University of Chinese Academy of Sciences, 100049 Beijing, China. Correspondence and requests for materials should be addressed to P.J. (email: pengjiang@dicp.ac.cn) or to X.B. (email: xhbao@dicp.ac.cn)

Low-cost and high-performance ultra-broadband photodetectors with spectral response from ultraviolet (UV) to long-wavelength infrared (LWIR) are highly desired in wide application scenarios[1,2]. Especially, photodetectors operating in the LWIR (8–14 μm) region, one important atmospheric window, play a key role in remote sensing and imaging. Photothermoelectric (PTE) effect, based on Seebeck effect, enables to convert temperature difference induced by the absorbed light to electric voltage. Compared to other photodetection mechanisms, i.e., photovoltaic, photoconductive and bolometric, the PTE effect can achieve ultra-broadband detection without external bias at room temperature (RT)[3]. Although the pyroelectric effect also possesses these two important features, an external chopper is required in the case of continuous-wave (CW) light detection[4]. Recently, the photo-induced thermo-phototronic effect[5] and pyro-phototronic effect[6,7] have been demonstrated to greatly enhance the performances of photodetectors.

The PTE effect of two dimensional (2D) materials, such as graphene[8–11], MoS₂[12], black phosphorus[13,14] and topological insulators[15,16] has drawn special attention with the rise of graphene in the last decade. In PTE effect, the responsivity is related to the light-induced temperature gradient and the material's Seebeck coefficient. The former is determined by the absorption and heat capacity, and the latter hinges on the density of states near the Fermi level[17]. The Seebeck coefficient of graphene[18] and black phosphorus[19] thin film at RT is generally tens of μV K⁻¹. In order to improve the responsivity, gate bias is employed to modulate the Fermi level to increase the Seebeck coefficient[8–10,14]. Furthermore, for photodetectors based on the mechanical exfoliated 2D materials, the effective active area is extremely small, thus highly precise illumination and costly microfabrication are needed to observe the photoresponse. In addition, due to the saturation effect, the 2D material-based PTE photodetectors usually can only be used to detect low-power light[8,11,13]. In order to build a larger temperature gradient to further improve the responsivity, plasmon-enhanced[20,21] and phonon-enhanced[22] photon absorptions have been applied successfully. For example, plasmon resonance of n-type Bi₂Te₃ and p-type Sb₂Te₃ wire array has been utilized to enhance the responsivity up to 38 V W⁻¹ in the visible region[20]. It will be of great significance to extend the spectral range to the LWIR region. Recently, the phonon absorption of SiO₂ substrate in the LWIR region is utilized to enhance the PTE response of graphene[22]. Nevertheless, due to the low intrinsic response of graphene, even the peak responsivity in the absorption band can only reach 78 nA W⁻¹, limiting its practical applications[2,23].

In SrTiO₃ (STO)—an important semiconductor oxide[24,25]—exotic physical phenomena related to its interface have been demonstrated[26–29]. STO has also been well studied as a high-temperature thermoelectric material due to its thermal and chemical stability[30,31]. The Seebeck coefficient of STO at RT can reach up to $-10^3$ μV K⁻¹, determined by its carrier density[32]. Furthermore, STO has a strong phonon absorption band in the LWIR atmospheric window, which could improve the photo-responsivity by increasing the temperature difference in this important spectral range[33]. Considering these two properties, excellent LWIR photodetection performance can be expected in STO-based PTE photodetector. It should be noted that there are a few works about STO single crystal photodetectors driven by off-diagonal thermoelectric effect[34]. This effect relies on the longitudinal temperature difference, perpendicular to the material surface. Under short-pulse laser illumination, it has relatively high responsivity, while for CW excitation, the responsivity is extremely low[35,36].

Herein we demonstrate the ultra-broadband response of reduced STO (SrTiO₃₋δ, r-STO) single crystal from 325 nm to 10.67 μm driven by the PTE effect. More importantly, phonon-enhanced photoresponse in the LWIR region is identified, and the responsivity at 10.67 μm is up to 1.2 V W⁻¹. Considering the existence of a phonon mode in STO at 2.62 THz, it is possible to further extend the spectral response to terahertz region[37]. Furthermore, r-STO photodetector exhibits linear response up to 1235 W cm⁻², a great advantage over 2D material-based photodetectors for detecting high-power light[8].

## Results

**Electronic and optical properties of SrTiO₃.** A commercial undoped STO single crystal is first cut into small pieces with 10 mm in length, 0.5 mm in width and 0.15 mm in thickness, then is further reduced under H₂ atmosphere at 900 °C for 4 h (see Methods for details). Fig. 1a shows the schematic diagram of the PTE effect measurement setup, and a photograph of the setup is provided in Supplementary Fig. 1. The left side of STO is glued onto a glass slide using double sided Kapton tape (thickness, 0.11 mm), while the other side is suspended. Two pairs of T-type thermocouples are anchored to the two ends of STO crystal using silver paint. In addition to recording the temperature, the Cu leads of the thermocouples are used to measure the photovoltage. Therefore, temperatures at each end and photovoltage across the sample can be measured in sequence. This configuration allows us to accurately understand the mechanism of STO ultra-broadband photodetectors.

We first examine the optical absorption of r-STO from UV to LWIR. The r-STO crystal exhibits a sharp absorption edge at around 400 nm, which corresponds to the onset of interband excitation (see Supplementary Fig. 2). A sub-band-gap absorption exists until the emergence of a broad absorption band in the LWIR region (Fig. 1b). This absorption band with the peak position located at 11.5 μm is related to the longitudinal optical phonon

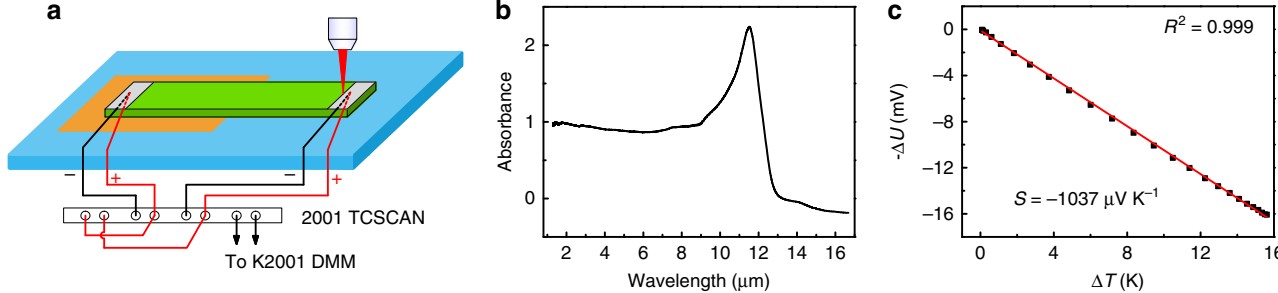

**Fig. 1** Electronic and optical properties of SrTiO₃. **a** Schematic measurement setup for the characterization of SrTiO₃ (STO) photodetector with half supported configuration. Keithley 2001 digital multimeter (DMM) equipped with a 2001-TCSCAN scanner card is utilized to record the photovoltage and the temperature variations. **b** Absorption spectrum of reduced STO (r-STO), which is annealed at 900 °C for 4 h under H₂ atmosphere. **c** Voltage across the r-STO versus the corresponding temperature difference to determine the room-temperature Seebeck coefficient (S) of r-STO. The red line is the linear fit to the experimental data. And the resulting coefficient of determination ($R^2$) is 0.999

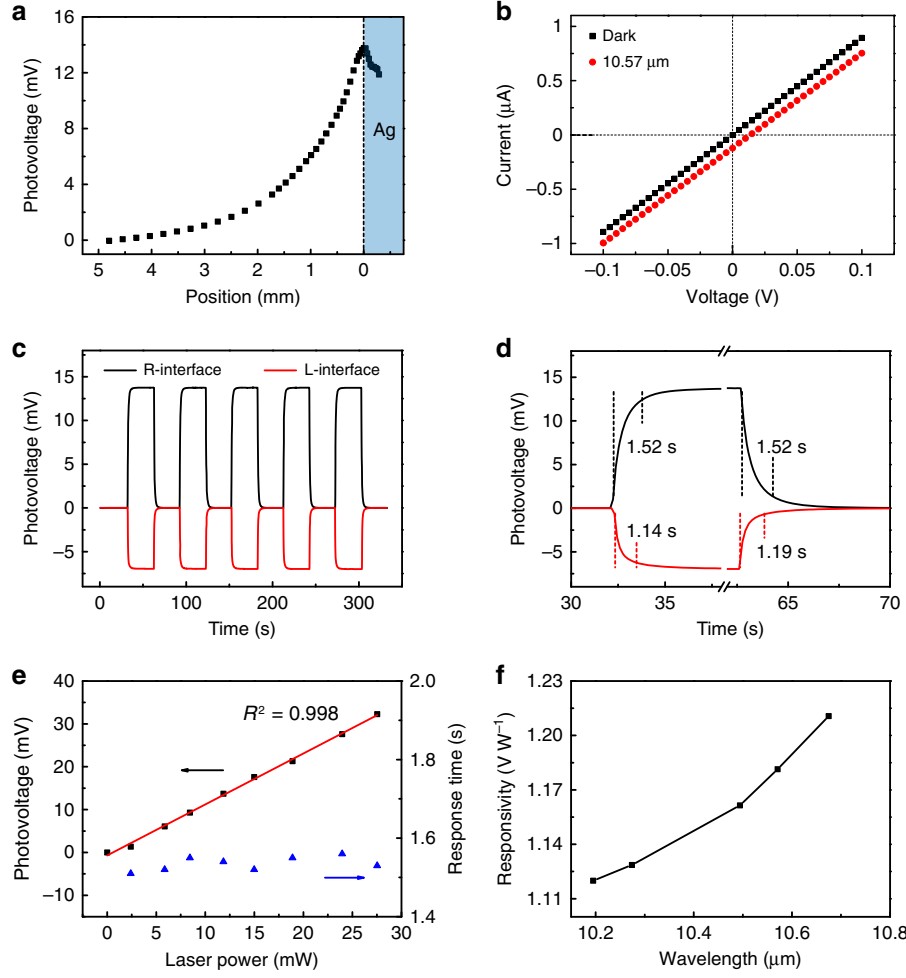

**Fig. 2** Long-wavelength infrared response. **a** Position-dependent photovoltage with the laser spot moved from the middle of STO to the right Ag electrode. The dashed line indicates the STO/Ag interface. **b** Current-voltage (*I-V*) curves in the dark and with laser illuminating the right interface. **c** Temporal response with the laser spot positioned on the right STO/Ag interface (R-interface) and left one (L-interface). **d** Enlargement of **c** to illustrate the response times. The rise/fall time is 1.52/1.52 s for the R-interface illumination, and 1.14/1.19 s for the L-interface illumination. **e** Dependence of photovoltage and response time on the laser power. The red line is the linear fit to the data, and obtained $R^2$ is 0.998. **f** Responsivity as a function of incident wavelength. The laser wavelength ($\lambda$) in panels **a–e** is 10.57 μm. The laser power in **a–d** is 11.6 mW. The illumination position in **e**, **f** is at the right STO/Ag interface. The laser spot is about 30 μm

mode[38]. Next, the transport properties of r-STO at RT are studied. The Seebeck coefficient (*S*) characterizes the ability of a material to convert a temperature difference ($\Delta T$) into an electrical voltage ($\Delta U$), defined as $S = -\Delta U/\Delta T$. The measurement of Seebeck coefficient is carried out in a probe station with the schematic shown in Supplementary Fig. 3. For STO reduced at 900 °C for 4 h, oxygen vacancies are produced, and the defect equation written in Kroger-Vink notation is $O_O^X = V_o^{\cdot\cdot} + 1/2O_2(g) + 2e'$, where $O_O^X$ represents oxygen ion sitting on an oxygen lattice site, $V_O^{\cdot\cdot}$ is an oxygen vacancy with +2 charge and $e'$ is an electron[39,40]. The resulting resistance and Seebeck coefficient is 113 kΩ and −1037 μV K$^{-1}$ (Fig. 1c), respectively. Dark *I-V* curve with the applied voltage ramping over a wide range (from 1 V to −1 V) demonstrates a perfect linear behavior (see Supplementary Fig. 4), indicating that an Ohmic contact, rather than Schottky contact, is formed between r-STO and silver paint[41].

**Photoresponse in the long-wavelength infrared region**. Figure 2a shows the illumination position-dependent photoresponse of r-STO with incident wavelength ($\lambda$) of 10.57 μm. The focused spot size of the laser beam is approximately 30 μm, and the laser

power (*P*) is ~11.6 mW. As the laser spot is swept from the middle of STO to the right electrode, the photovoltage monotonically increases from nearly zero. The maximum photovoltage occurs at the STO/electrode interface. Next, *I-V* curve with laser illuminating on the right STO/electrode interface (denoted as R-interface, and L-interface for the left STO/electrode interface) is measured (red dotted line in Fig. 2b). Compared to the dark *I-V* curve (black dotted line in Fig. 2b), the illuminated *I-V* curve shifts downward with no evident change in the extracted resistance. These phenomena are the typical characteristics of PTE effect. Fig. 2c presents the temporal photovoltage ($V_{ph} = V_{light} - V_{dark}$) responses with laser illuminating the R-interface and L-interface, both of which reveal excellent on-off repetition. As the illumination position is switched from the R-interface to the L-interface, the corresponding responsivity ($R = V_{ph}/P$) decreases by 50% from 1.18 to 0.59 V W$^{-1}$. The higher sensitivity of R-interface originates from the larger temperature increase due to the suspended structure with less thermal loss to the substrate. Fig. 2d presents the photovoltage response time, defined as 10–90% signal change. The rise/fall time ($\tau_r/\tau_f$) is determined to be 1.52/1.52 s for the R-interface illumination, and 1.14/1.19 s for the L-interface illumination, much faster than that of Thorlab

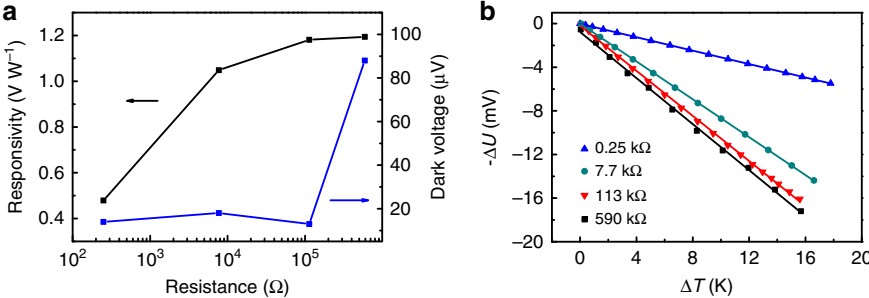

**Fig. 3** Effects of SrTiO₃ internal resistance. **a** Responsivity and dark voltage as a function of STO resistance at $\lambda = 10.57\,\mu m$. **b** Dependences of the output voltage on the temperature difference used to extract Seebeck coefficients of STO crystals with resistance of 0.25, 7.7, 113 and 590 kΩ. The corresponding Seebeck coefficients are −307, −867, −1037, −1055 μV K⁻¹, respectively

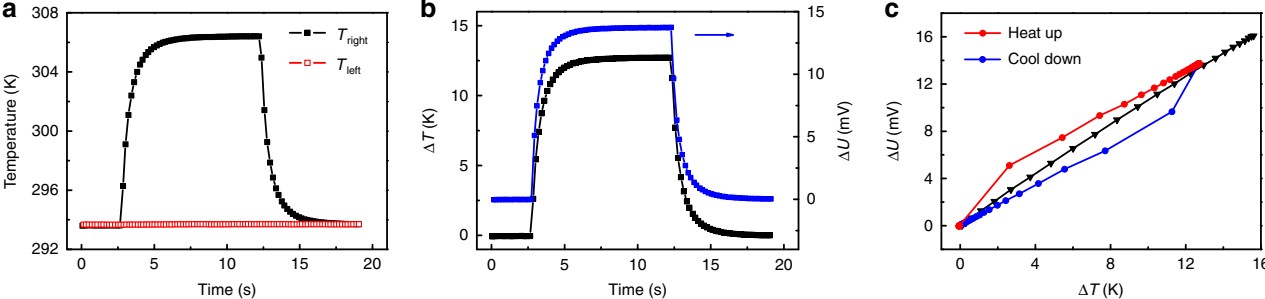

**Fig. 4** Photoresponse mechanism. **a** Temperature response of the two ends of r-STO crystal to the laser illumination (10.57 μm) on the right STO/Ag interface. The laser power is about 11.6 mW, and the spot size is about 30 μm. **b** Time-dependent temperature difference and the corresponding photogenerated voltage across the STO crystal. The dark voltage has not been subtracted. **c** Photogenerated voltage versus the temperature difference deduced from **b**. The black data points are taken from Fig. 1d for comparison, in which case the temperature difference is built up by the external resistance heating

S175C power detector (~40 s, measured at the same conditions) used for calibration. As a result, the half-suspended configuration allows the significant improvement of responsivity to be achieved at the expense of a slight increase in the response time.

In the following, we mainly focus on the photoresponse properties with the laser spot illuminating at the R-interface. Supplementary Fig. 5 shows the time-dependent photovoltage response under excitation of different laser powers. Raising the laser power to 27.5 mW, steady-state photovoltage still can be obtained. The photovoltage increases linearly with the laser power, while the response time remains almost unchanged (Fig. 2e). We further investigate the dependence of responsivity on the illumination wavelength between 10.19 μm to 10.67 μm (Fig. 2f). It is evident that the responsivity increases with the incident wavelength. This dependency is consistent with the absorption spectrum of STO (Fig. 1b), suggesting phonon-enhanced absorption can effectively improve the photoresponsivity.

The internal resistance of STO has a significant impact on the photodetector performance. Three STO samples with different resistance (590, 113 and 7.7 kΩ) are prepared by annealing undoped STO at 900 °C for 3, 4 and 16 h, respectively, while the sample with a resistance of 250 Ω is the commercial 0.7 weight% Nb-doped STO. As the internal resistance increases from 250 Ω to 590 kΩ, the responsivity firstly increases quickly and then tends to approach saturation (Fig. 3a). The increase of responsivity with resistance originates from the increase of the Seebeck coefficient (Fig. 3b). When the resistance is in the range of 250 Ω to 113 kΩ, the measured dark voltages are less than 20 μV. By further increasing the resistance to 590 kΩ, the dark voltage increases dramatically, and is more susceptible to electromagnetic interference. The response times of these samples are all on the order of 1 s. Considering these factors, the ideal internal reisistance is around 100 kΩ to achieve high responsivity and low dark voltage.

We further examine the the effect of r-STO channel length on the photodetector performance (see Supplementary Fig. 6). As the length of r-STO decreases from 10 mm to 3.1 mm, the responsivity and response time decrease to 0.82 V W⁻¹ and 0.93 s, repetively. This trend is consistent with that of carbon nanotube PTE detectors[42], due to the larger temperature gradient for longer channel length. Therefore, high-speed STO photodetector can be obtained at the expense of responsivity.

**Photoresponse mechanism.** To further understand the mechanism behind the PTE response in r-STO, we simultaneously measure the temperature rise and the corresponding photovoltage with the laser ($\lambda = 10.57\,\mu m$, $P = 11.6$ mW) positioned at the R-interface. Fig. 4a shows the temperature evolutions of the two ends with laser switched on and off. Interestingly, the temperature of the right end increases by 12.7 K, while that of the left end remains almost unchanged, regardless of the relatively high thermal conductivity of STO (~8 W m⁻¹ K⁻¹)[43]. As comparision, for fully supported configuration, the temperature difference is only about 7.9 K (see Supplementary Fig. 7). These phenomena demonstrate that the half supported configuration is beneficial to establish a larger temperature gradient.

In Fig. 4b, we compare the time-dependent temperature difference (ΔT) and the corresponding voltage (ΔU, dark voltage is not subtracted) between the two ends. Although they exhibit similar changing trends, a distinction still can be identified by plotting ΔU versus ΔT (Fig. 4c). Compared to the nearly perfect linear relationship when external resistance heating is employed to build up a temperature gradient (Black line in Fig. 4c), ΔU shows an abrupt increase when the laser is turned on, then increases linearly with ΔT. When the laser is turned off, ΔU shows an abrupt decrease, then decreases linearly with ΔT,

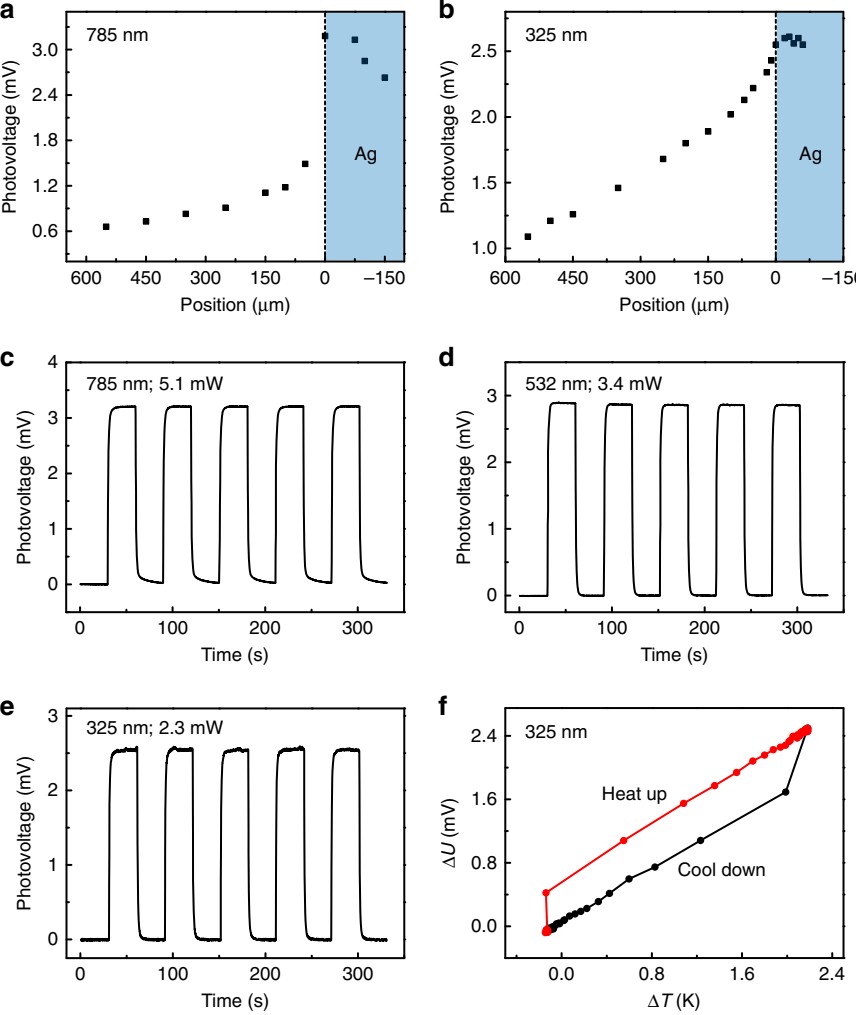

**Fig. 5** Ultraviolet, visible and near infrared response. **a**, **b** Position-dependent photoresponses around the right STO/Ag interface at **a** $\lambda = 785$ nm with power of 5.1 mW and **b** 325 nm with power of 2.3 mW. **c–e** Temporal responses at **c** $\lambda = 785$ nm, **d** 532 nm and **e** 325 nm. **f** Photogenerated voltage versus the temperature difference at $\lambda = 325$ nm. The illumination position in panels **c–f** is at the right STO/Ag interface. The spot sizes of the 785 nm, 532 nm and 325 nm lasers are about 10 μm, 6 μm and 5 μm, respectively

forming a hysteresis-like trace. This hysteresis-like behavior still exists even when the laser spot is moved to pure r-STO region, 500 μm away from R-interface (see Supplementary Fig. 8), indicating this phenomenon is not related to the STO/Ag interface. We further investigate the PTE response of another bulk material, i.e., vanadium-doped $MoS_2$ with thickness of about 400 μm, and also observe similar hysteresis-like trace (see Supplementary Fig. 9). We propose that this hysteresis might originate from the bulk nature of the measured materials. Considering the thickness of r-STO (~150 μm), the surface temperature is different from the bottom temperature right after laser illumination, and there exists thermal diffusion perpendicular to the r-STO surface, which is a common phenomenon in photothermal effect of bulk materials[44]. The photoresponse from surface heating approaches the steady state in a short time, and the contributed photovoltage can be estimated from the vertical distance between the red and blue lines in Fig. 4c (i.e., ~2.8 mV). Establishing thermal equilibrium in the whole sample requires longer time. When this equilibrium is achieved, the resulting data point ($\Delta T = 12.7$ K, $\Delta U = 13.7$ mV) is almost on the black line of $\Delta U$ versus $\Delta T$ under external resistance heating (Fig. 4c). Compared with the laser heating, the heating power under external resistance heating is much larger, so there is no lag

between the surface and bottom temperature, and the correlation between $\Delta U$ and $\Delta T$ is linear and crosses the zero point.

**Broadband photoresponse**. Figure 5 displays the ultra-broadband spectral response of r-STO. Fig. 5a, b show the position dependent photovoltage response under 785 nm (laser spot, ~10 μm; power, 5.1 mW) and 325 nm laser illumination (laser spot, ~5 μm; power, 2.3 mW), respectively. For the near-IR (NIR) illumination ($\lambda = 785$ nm) on r-STO, photovoltage signal still exists due to the sub-band-gap absorption. Compared to the gradual increase in photovoltage when moving the laser spot towards the STO/Ag interface under LWIR (Fig. 2a) and UV (Fig. 5b) illumination, the photovoltage under NIR illumination exhibits an abrupt increase (Fig. 5a), indicating the absorption of NIR light by Ag electrode contributes a lot to the photoresponse at the interface. Fig. 5c, d and e show the on/off response with the laser positioned at the R-interface at $\lambda = 785$ nm, 532 nm, and 325 nm. Excellent switching behavior can be observed for all cases. The resulting responsivities are 0.63, 0.85 and 1.11 V W$^{-1}$, respectively, as summarized in Fig. 6a. The dependence of $\Delta U$ on $\Delta T$ at $\lambda = 325$ nm in Fig. 5f indicates that, even when the incident photon energy is larger than the band gap of STO, PTE

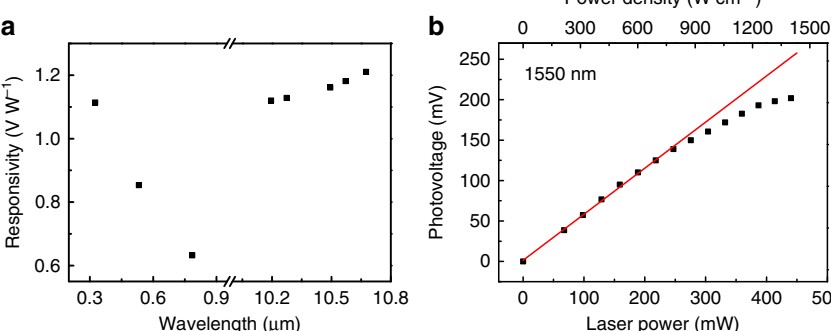

**Fig. 6** Ultra-broadband response and high-power-density detection. **a** Summary of the spectral response of r-STO photodetector. **b** Photovoltage versus laser power of r-STO detector at $\lambda = 1550$ nm, exhibiting linear response up to 246 mW (786 W cm$^{-2}$). The red line is the linear fit to the data from 0 to 246 mW. The spot size of the 1550 nm laser is about 200 μm

mechanism still works, not photovoltaic effect. For the photovoltaic effect, a Schottky contact is required to separate photo-generated electron-hole pairs in the case of metal-semiconductor-metal configuration[45], whereas the contact between Ag electrode and r-STO is an Ohmic contact, which is corroborated by the linear I-V curve shown in Supplementary Fig. 4.

Lastly, considering the stability of STO, we explore the response of r-STO photodetector under high-power-density light illumination. The measurements are performed on another independent platform equipped with 1550 nm laser, to avoid the damage to the optical elements of Raman and near-field optical microscopies used in previous discussions. Fig. 6b presents the dependence of the photovoltage on the laser power at $\lambda = 1550$ nm with laser spot of about 200 μm. The photodetector exhibits linear response up to 246 mW (786 W cm$^{-2}$) with the responsivity of 0.57 V W$^{-1}$. With further increasing the incident power, the output voltage tends to be saturated. It is worth noting that the absolute Seebeck coefficient of STO increases slightly with temperature[43]. Considering that the responsivity of the PTE photodetector is proportional to the Seebeck coefficient and the temperature difference, this saturation effect at high power density is attributed to the saturated temperature difference. Under high-power light illumination, the heat, which is transfered from the suspended hot end to non-suspended cold end, cannot dissipate immediately. Therefore the non-suspended end will also be heated up, resulting in the saturation of effective temperature gradient. The response linearity can be further improved by replacing the half-supported configuration with the fully supported one at the cost of responsivity (see Supplementary Fig. 10). The r-STO detector with fully supported configuration has a linear response up to 388 mW (1235 W cm$^{-2}$) with the responsivity reduced to 0.35 V W$^{-1}$. In previous graphene PTE photodetector, the linear response can only maintain up to 20 μW[8]. The ability of r-STO photodetector to detect low-power radiation is investigated as well (see Supplementary Fig. 11). When a human finger is placed above the suspended end with a separation of about 1.5 mm, the output voltage can be up to 0.93 mV. The sensitivity of STO photodetector on human radiation is much higher than graphene-based PTE detector, which only generates the photovoltage on the order of nV[46].

## Discussion

We have demonstrated the ultra-broadband response of r-STO photodetectors based on the PTE effect at room temperature. The proposed photodetector is very facile to prepare, which is crucial for realistic applications. The responsivity of a single-unit r-STO photodetector is comparable to, or even better than 2D materials-based and commercial PTE photodetectors, which usually require complicated microfabrication processes (see Supplementary

Table 1). Moreover, the ability of operation under high-power density illumination renders r-STO a unique advantage over 2D materials. We further reveal that the existence of a Ti–O phonon mode could enhance the photoresponse in the LWIR region, which provides an effective scheme to improve the sensitivity of photodetectors. Further improvement of STO-based photodetector can be envisioned by fabricating STO thin film to reduce the response time and utilizing nanophotonics to enhance the responsivity[20,21].

## Methods

**SrTiO$_3$ treatment**. A commercial undoped STO (100) single crystal (Hefei Kejing Materials Technology Co., Ltd., China) is cut into small pieces with 10 mm in length, 0.5 mm in width and 0.15 mm in thickness. Then, the STO crystal is reduced in a low-pressure horizontal furnace, which is evacuated to a base pressure of about 1 Pa, and then purged with Ar for several times before heating. The furnace is heated up to 900 °C at a rate of 7.5 °C min$^{-1}$ with H$_2$ flow rate of 70 sccm, held at 900 °C for 4 h, then cooled to room temperature.

**Photoresponse measurement**. The photoresponse in the LWIR region is measured using a CO$_2$ gas laser (Access laser), which is focused by a parabolic mirror. For the measurement in the NIR and visible spectral range, 785 nm and 532 nm lasers in Bruker SENTERRA Raman confocal microscope are used. Horiba HR800 Raman microscope with an external 325 nm laser (Model: IK3301R-G, Kimmon Koha Co. Ltd.) is employed to measure the UV response. The incident laser powers are calibrated using Thorlab S175C thermal sensor. I-V curve and photovoltage are measured with Keithley 2450 source meter. Keithley 2001 digital multimeter (DMM) equipped with a 2001-TCSCAN scanner card is utilized to record the photovoltage, as well as the temperature variations. For the high-power-density illumination, a power-tunable 1550 nm solid-state laser is used with the focused laser spot of about 200 μm, and the STO detector is mounted on a manual translation stage to optimize the illumination position.

**Electronic and optical properties characterizations**. The ultraviolet-visible-near infrared spectrum of r-STO is obtained with PerkinElmer Lambda 950 spectrophotometer. The fourier transform infrared (FTIR) spectrum is taken with Bruker Hyperion 3000 FTIR microscope. The measurement of Seebeck coefficient is carried out using the Lakeshore TTPX probe station equipped with modified sample stage and Keithley 2001 DMM.

## Data availability

The data that support the findings of this study are available from the corresponding authors on request.

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

## Acknowledgements

P.J. acknowledge the financial support from the National Key Research and Development Program of China (Grant No. 2016YFA0203500) and Dalian Institute of Chemical Physics (Grant No. DICP ZZBS201608).

## Author contributions

P.J. and X.H.B. conceived the original concept. X.W.L. performed the experiment. P.J. and X.W.L. analyzed the data and wrote the paper.

## Additional information

**Competing interests:** The authors declare no competing interests.

