## [Peer Review File · Nature Communications]

Reviewers' comments:

Reviewer #1 (Remarks to the Author):

The authors demonstrated a reduced SrTiO₃ (r-STO)-based PTE photodetector with high sensitivity to 1.2 V/W and broadband spectral response from the 325 nm to 10.67 μm at room temperature. The photodetector has simple structure, preparation method and high performance. The semi-support configuration allows the significant improvement of responsivity to be achieved at the expense of a slight increase in response time. The outstanding performance of STO PTE-photodetector is attributed to its intrinsic high Seebeck coefficient and strong phonon-enhanced photoresponse in the long wavelength infrared region. The ability of operation under high-power density illumination renders r-STO great advantage over 2D materials. Further improvement of STO-based photodetector can be envisioned by fabricating STO thin film with finer structure. The manuscript presents an interesting research work. However, there are still a couple of issues needed to be noted and it can be published after major revisions. The detailed comments are as follows according to the order of the manuscript:

1. In Page 1 (Line 13-14), about this sentence 'Previous PTE-photodectors usually employ traditional good thermoelectric materials with Seebeck coefficients in the range of 100 μV/K, which restricts the sensitivity', which maybe not quite appropriate, how you can make sure the range of Seebeck coefficients above restricts the sensitivity of PTE-photodectors. It seems that there is no inevitable relationship between them, it seems that materials with small Seebeck coefficient may have high sensitivity, vice versa.
2. In Page 3 (Line 29-32), about several photodetection mechanisms, the authors should cite the related published papers in the introduction section and add some description (Ya Yang's research group has firstly developed the new strategies of the photo-induced thermo-phototronic effect and pyro-phototronic effect to realize the impressed light detection and largely enhance the performances of photodetectors, respectively.) about these cited papers (1. Photocurrent Polarity Controlled by Light Wavelength in Self-Powered ZnO Nanowires/SnS Photodetector System, iScience 1, 16–23, March 23, 2018; 2. Photovoltaic–Pyroelectric Coupled Effect Induced Electricity for Self-Powered Photodetector System, Adv. Mater. 2017, 1703694; 3. Enhanced P3HT/ZnO Nanowire Array Solar Cells by Pyro-phototronic Effect, ACS Nano 2016, 10, 10331–10338; 5.)
3. In Page 4 (Line 70-72) and Page 5 (Line 98-99), the authors should refer to some published papers about these sentences respectively.
4. In Page 5, Fig. 1a, the fabricated process and three demensions of semi-support configuration of photodetection device should be descriptive specifically.
5. In Page 6 (Fig. 2a), the unit of the position should be 'mm' rather than 'μm'. In Page 7 (Line 128-129), the preparation process of the specimens annealed at 900°C for different time should be described specifically. Additionally, the data of Fig. 2g-h were a little small, it is suggested that some additional points be added.
6. In Page 7 (Line 143-147), the authors just investigated the dependence of responsivity on the illumination wavelength between 10.19 μm to 10.67 μm, according to Fig. 1b, the absorption band with the peak position was located at 11.5 μm, so why not research the wavelength around 11.5 μm ? It is recommended the sensitivities of r-STO photodetector with wavelength more than 11.5 μm should be added.
7. In Page 8 (Line 158), how the relatively high thermal conductivity of STO (~8 W m⁻¹ K⁻¹) was calculated or obtained. Here, it should be explained or refers to some papers.
8. In Page 9-10 (Line 197-209), "The resulting responsivities are 1.11, 0.85 and 0.63 V/W, respectively", about this sentence, please adjust the right order of three responsivities correspond to the preceding sentence. In addition, laser spots and powers of the three laser illumination are all different, whether consider keep the same laser spot or power. Lastly, how you can make sure the absorption of NIR light by Ag electrode contributes a lot to the response at the interface (Line 201-204). Can you make a contrastive experiment.
9. In Page 10 (Line 213), the authors said 'explored the response of r-STO photodetector under high-power-density light illumination', why did you choose the wavelength of 1550 nm and the laser spot of about 200 μm? Which were different from above suddenly, please add some

explanations.

Reviewer #2 (Remarks to the Author):

In the manuscript "Phonon-enhanced photothermoelectric effect in SrTiO₃ ultra-broadband photodetector" Xiaowei Lu and co-authors present an interesting work where a ultra-broadband photodetector based in STO is fabricated and characterized. The device exploits the photothermoelectric effect to obtain the ultra-broadband sensitivity. The STO has a nice combination of large Seebeck effect with a sub-bandgap strong absorption (due to phonon-enhanced absorption in the LWIR band). The manuscript is rather insightful and it seems that the authors have done many double-checks in order to be sure of the origin of the observed photocurrent.

The manuscript is well-written (the text can be easily followed) and the authors explained well all the different measurements that carried out and why they do them. I think that a more thorough discussion between the state-of-the-art photodetectors in that long wavelength range and the current photodetector will be still needed. The authors did compare their results with a commercial photodetector but a more thorough comparison is really crucial.

I think that the article is clearly interesting and it would be worth of publication in Nature Communications but it could be further improved.

- 1) As a reader, I missed actual pictures of the fabricated devices instead of the cartoons.
- 2) As pointed out above, a more thorough comparison with different photodetectors will be crucial.
- 3) The authors showed an absorption below 400 nm and then when they showed the photoresponse upon 325 nm illumination it seems that it is all due to photothermoelectric as well. How can it be that for above-bandgap illumination photovoltaic/photoconductive is not the most dominant photocurrent generation mechanism.
- 4) The authors didn't discuss what are the processes that explain the sub-linear relationship of photovoltage and power (figure 5b) for high power illumination.
- 5) It would be nice for the readership to show the actual traces used to measure the Seebeck coefficient of the different STO substrates (with different resistances) in the supporting information.
- 6) The discussion about the two different decay times started giving a lot of importance to the hot carriers and it could be misleading for the readership. They could reduce the discussion saying that hot carriers are clearly ruled out as they relax in a completely different time scale. The two timescales are actually not so different. How bad is the exponential fitting using a single exponential?
- 7) The photodetector is rather big and upon illumination the temperature increase might be substantial (even more than 10K). I would strongly recommend using a thermal camera imaging to map the temperature distribution in their devices. This would be very illustrative when comparing the device that is partially overhanging and the one that lays flat on the substrate.
- 8) Figure 2 is particularly crowded. While Reading the manuscript I was going back and forth to Figure 2 all the time which makes the reading more cumbersome. Maybe the authors could

consider splitting the figure in two.

Reviewer #3 (Remarks to the Author):

Dear authors,

Several questions arose during the review of your paper "Phonon-enhanced photothermoelectric effect in SrTiO₃ ultra-broadband photodetector".

You name your device/paper a photodetector. I strongly disagree with that and believe it is an overstatement; it does not represent an electronic device but a material with silver paint contacts.

You mention e.g. night vision and sensing as potential applications. However, you use rather high laser powers and do not demonstrate the possibility to detect low intensity light contrary to those claims. Can you experimentally demonstrate the suitability of your detector for low light intensity applications?

You claim an ultra-broad spectral response from 325nm-10.67um; however there is a large gap from 1550nm-10um in your data. The claim is an overstatement without providing experimental evidence.

What is your assumption to detect THz radiation based on? No references, etc given.

You generally use graphene and 2D-materials as a comparison for the performance of your device. However, in line 75 you chose nanoporous Si as reference to underline the "high" speed of your device and omit graphene and 2D-materials. Instead of cherry picking, please also add the performance of graphene and 2D-materials to fairly compare your device performance. In line 95, you claim annealing leads to oxygen vacancies. However, neither experimental evidence nor a reference is provided.

Section photoresponse mechanism: You neither make a clear statement what the mechanism is nor do you present any convincing experimental data or theoretical modelling. The only statement is "a fast and slow component" and many guesses.

Overall, the manuscript presents rather a collection and description of experimental data without clear explanation. It suffers from overstatements and lacks relevant comparisons and references. Further, the English language needs to be improved.

Reply to reviewers' comments: NCOMMS-18-08247-T

Reviewer #1:

The authors demonstrated a reduced SrTiO₃ (r-STO)-based PTE photodetector with high sensitivity to 1.2 V/W and broadband spectral response from the 325 nm to 10.67 μm at room temperature. The photodetector has simple structure, preparation method and high performance. The semi-support configuration allows the significant improvement of responsivity to be achieved at the expense of a slight increase in response time. The outstanding performance of STO PTE-photodetector is attributed to its intrinsic high Seebeck coefficient and strong phonon-enhanced photoresponse in the long wavelength infrared region. The ability of operation under high-power density illumination renders r-STO great advantage over 2D materials. Further improvement of STO-based photodetector can be envisioned by fabricating STO thin film with finer structure. The manuscript presents an interesting research work. However, there are still a couple of issues needed to be noted and it can be published after major revisions.

We thank the referee for your careful reading of the manuscript and helpful comments.

Comment 1. In Page 1 (Line 13-14), about this sentence ‘Previous PTE-photodectors usually employ traditional good thermoelectric materials with Seebeck coefficients in the range of 100 μV/K, which restricts the sensitivity’ , which maybe not quite appropriate, how you can make sure the range of Seebeck coefficients above restricts the sensitivity of PTE-photodectors. It seems that there is no inevitable relationship between them, it seems that materials with small Seebeck coefficient may have high sensitivity, vice versa.

Following the advice of the referee, we have deleted previous description, and replaced it with “and array structure with multiple units is usually employed to enhance the photodetection performance”. (Page 1, Line 14-15).

Comment 2. In Page 3 (Line 29-32), about several photodetection mechanisms, the authors should cite the related published papers in the introduction section and add some description (Ya Yang's research group has firstly developed the photo-induced thermo-phototronic effect and pyro-phototronic effect to realize the impressed light detection and largely enhance the performances of photodetectors, respectively.) about these cited papers (1. Photocurrent Polarity Controlled by Light Wavelength in Self-Powered ZnO Nanowires/SnS Photodetector System, iScience 1, 16–23, March 23, 2018; 2. Photovoltaic–Pyroelectric Coupled Effect Induced Electricity for Self-Powered Photodetector System, Adv. Mater. 2017, 1703694; 3. Enhanced P3HT/ZnO Nanowire Array Solar Cells by Pyro-phototronic Effect, ACS Nano 2016, 10, 10331–10338; 5.)

The recommended photodetection mechanisms are meaningful, and we have cited the relevant papers in the revised version. (Page 2, Line 33-36)

Comment 3. In Page 4 (Line 70-72) and Page 5 (Line 98-99), the authors should refer to some published papers about these sentences respectively.

The reference indicating the existence of STO phonon mode in the terahertz region has been added. (Page 3, Line 74-75)

The reference about Ohmic and Schottky contacts has been added. (Page 5, Line 104-105)

Comment 4. In Page 5, Fig. 1a, the fabricated process and three demensions of semi-support configuration of photodetection device should be descriptive specifically.

We have added some description in Page 4 (Line 79-84), and more detailed information is included in Methods section.

Comment 5. In Page 6 (Fig. 2a), the unit of the position should be “mm” rather than “ μm ”. In Page 7 (Line 128-129), the preparation process of the specimens annealed at 90 °C for different time should be described specifically. Additionally, the data of Fig.

2g-h were a little small, it is suggested that some additional points be added.

The unit of the horizontal axis in Figure 2a has been corrected.

The annealing conditions have been given in the revised manuscript (Page 6, Line 138-140). “Three STO samples with different resistance (590, 113 and 7.7 k Ω) were prepared by annealing undoped STO at 900 °C for 3, 4 and 16 h, respectively, while the sample with a resistance of 250 Ω is the commercial 0.7 wt% Nb-doped STO.”

The data points in previous Figure 2g,h (currently Figure 3a,c) have been enlarged.

Comment 6. In Page 7 (Line 143-147), the authors just investigated the dependence of responsivity on the illumination wavelength between 10.19 μm to 10.67 μm , according to Fig. 1b, the absorption band with the peak position was located at 11.5 μm , so why not research the wavelength around 11.5 μm ? It is recommended the sensitivities of r-STO photodetector with wavelength more than 11.5 μm should be added.

Because the maximum output wavelength of the used CO₂ laser is 10.7 μm , we currently cannot measure the optical response around 11.5 μm . When the necessary experimental condition is accessible, we will further measure this property.

Comment 7. In Page 8 (Line 158), how the relatively high thermal conductivity of STO ($\sim 8 \text{ W m}^{-1} \text{ K}^{-1}$) was calculated or obtained. Here, it should be explained or refers to some papers.

We have added the relevant reference about the thermal conductivity of STO. (Page 7, Line 159-160)

Comment 8. In Page 9-10 (Line 197-209), "The resulting responsivities are 1.11, 0.85 and 0.63 V/W, respectively", about this sentence, please adjust the right order of three responsivities correspond to the preceding sentence. In addition, laser spots and powers of the three laser illumination are all different, whether consider keep the

same laser spot or power. Lastly, how you can make sure the contributes a lot to the response at the interface (Line 201-204). Can you make a contrastive experiment.

The order of the three responsivities at $\lambda = 785, 325$ and 532 nm has been corrected. (Page 8, Line 200-201)

The measurements at $\lambda = 785, 532$ and 325 nm are conducted on a confocal Raman microscopy in which the laser spot size depends on the incident wavelength and used objective lens. Thus, we can only make sure the laser spot sizes are on the same order of magnitude.

Because of the linear dependence of photovoltage on laser power (Fig. 2e), after normalization, the difference in the illuminating power would have no impact on the quantitative analysis.

By comparing the illumination position-dependent photovoltage responses at 785 nm (Fig. 5a), 325 nm (Fig. 5b) and $10.57 \mu\text{m}$ (Fig. 2a), we make the conclusion that the absorption by Ag electrode contributes a lot to the photoresponse when STO/Ag interface is illuminated by 785 nm laser.

As shown from these three figures, only at $\lambda = 785$ nm, there exists an abrupt increase in photovoltage when the laser spot is moved from STO to STO/Ag interface. This phenomenon can be used to support our opinion. (Page 8, Line 191-195)

Comment 9. In Page 10 (Line 213), the authors said ‘explored the response of r-STO photodetector under high-power-density light illumination’, why did you choose the wavelength of 1550 nm and the laser spot of about $200 \mu\text{m}$? Which were different from above suddenly, please add some explanations.

To protect the optical elements of Raman microscopies and scanning near-field

optical microscopy used for photoresponse measurements at $\lambda = 785$ nm, 532 nm and 325 nm and in the long-wavelength infrared region, we used an independent platform equipped with 1550 nm high-power laser to explore the ability of STO detector to detect high-power-density radiation.

In the revised manuscript, we have added relevant explanation. (Page 8, Line 205-208)

Reviewer #2:

In the manuscript "Phonon-enhanced photothermoelectric effect in SrTiO₃ ultra-broadband photodetector" Xiaowei Lu and co-authors present an interesting work where a ultra-broadband photodetector based in STO is fabricated and characterized. The device exploits the photothermoelectric effect to obtain the ultra-broadband sensitivity. The STO has a nice combination of large Seebeck effect with a sub-bandgap strong absorption (due to phonon-enhanced absorption in the LWIR band). The manuscript is rather insightful and it seems that the authors have done many double-checks in order to be sure of the origin of the observed photocurrent.

The manuscript is well-written (the text can be easily followed) and the authors explained well all the different measurements that carried out and why they do them. I think that a more thorough discussion between the state-of-the-art photodetectors in that long wavelength range and the current photodetector will be still needed. The authors did compare their results with a commercial photodetector but a more thorough comparison is really crucial.

I think that the article is clearly interesting and it would be worth of publication in Nature Communications but it could be further improved.

We firstly thank the referee for your constructive comments on our manuscript.

Comment 1. As a reader, I missed actual pictures of the fabricated devices instead of the cartoons.

Following the referee's advice, the real photo of STO photodetector has been added. (Page 4, Line 82-83, Supplementary Figure 1)

Comment 2. As pointed out above, a more thorough comparison with different photodetectors will be crucial.

We have added a more thorough comparison between our STO photodetector and 2D materials-based photothermoelectric detectors. (Page 9, Line 230-234, Supplementary Table 1)

Comment 3. The authors showed an absorption below 400 nm and then when they showed the photoresponse upon 325 nm illumination it seems that it is all due to photothermoelectric as well. How can it be that for above-bandgap illumination photovoltaic/photoconductive is not the most dominant photocurrent generation mechanism.

Because our device is self-powered, and no external power source is applied, photoconductive mechanism can be ruled out.

As for photovoltaic mechanism, in the case of metal-semiconductor-metal configuration, the existence of Schottky junction is necessary to separate photogenerated electron-hole pairs. However, in our case, the contact between Ag and STO is Ohmic contact, not Schottky contact (Figure 2b). Consequently, although the photon energy of 325 nm laser is larger than the band gap of STO, the photothermoelectric mechanism still works.

In the revised manuscript, we have added relevant discussion. (Page 8, Line 198-203)

Comment 4. The authors didn't discuss what are the processes that explain the sub-linear relationship of photovoltage and power (figure 5b) for high power illumination.

In our half supported configuration, the glass slide can serve as heat sink. Under low power light illumination, the temperature of the cold side of STO detector doesn't change (Figure 4a). However, under high power light illumination, due to the large heat flux, the non-suspended end will also be heated, resulting in the saturation of effective temperature gradient.

In the photothermoelectric effect, the photovoltage is equal to the product of temperature gradient and Seebeck coefficient. The absolute Seebeck coefficient of STO usually increases slightly with temperature. Based on these, the nonlinear relationship in Figure 6b is due to the saturation of effective temperature gradient under high power light illumination.

In the revised manuscript, we have added relevant discussion. (Page 9, Line 210-217)

Comment 5. It would be nice for the readership to show the actual traces used to measure the Seebeck coefficient of the different STO substrates (with different resistances) in the supporting information.

Following the referee's advice, we have added the actual traces used to determine the Seebeck coefficients of STO with different resistance (Page 6, Line 142-143, Figure 3b).

Comment 6. The discussion about the two different decay times started giving a lot of importance to the hot carriers and it could be misleading for the readership. They could reduce the discussion saying that hot carriers are clearly ruled out as they relax in a completely different time scale. The two timescales are actually not so different. How bad is the exponential fitting using a single exponential?

In the revised manuscript, we have deleted the discussion about the hot-carrier assisted photothermoelectric effect.

We have provided the mono-exponential fitting for comparison (See the picture below). The obtained coefficient of determination is 0.9932, just slightly worse than that of the biexponential fitting (0.9999). In our original manuscript, we used bi-exponential fitting to quantitatively explain the hysteresis behavior under laser heating (Figure 4c). To make the whole story more readable without influencing the major conclusions, we chose to delete these discussions in current version.

Comment 7. The photodetector is rather big and upon illumination the temperature increase might be substantial (even more than 10K). I would strongly recommend using a thermal camera imaging to map the temperature distribution in their devices. This would be very illustrative when comparing the device that is partially overhanging and the one that lays flat on the substrate.

The photodetection characterizations were mainly conducted on Raman and scanning near-field microscopy platforms. Unfortunately, there is not enough room in these setups for thermal camera. Nevertheless, with the aid of Keithley 2001 multimeter, we can simultaneously measure the temperature changes of the hot and cold sides, as well as the photovoltage.

In the revised manuscript, we have added the relevant characterization of fully supported STO photodetector for comparison (Page 7, Line 161-162). As can be seen from above figures, when STO fully lies on the glass slide, the temperature of the hot end increases by 7.9 K, which is 4.8 K lower than the case of half supported configuration. This is due to part of the heat energy of the hot end is transferred to the underlying glass slide.

Comment 8. Figure 2 is particularly crowded. While Reading the manuscript I was going back and forth to Figure 2 all the time which makes the reading more cumbersome. Maybe the authors could consider splitting the figure in two.

Following the referee’s advice, we have split Figure 2 into two figures.

Reviewer #3:

Several questions arose during the review of your paper “Phonon-enhanced

photothermoelectric effect in SrTiO₃ ultra-broadband photodetector”.

We firstly thank the referee for the constructive comments which help to improve the quality of the paper.

Comment 1. You name your device/paper a photodetector. I strongly disagree with that and believe it is an overstatement; it does not represent an electronic device but a material with silver paint contacts.

Although the structure of our device is relatively simple, it indeed has the ability to detect light radiations. And in terms of photothermoelectric (PTE) effect, it is generally accepted that such kind of simple configuration is employed to make photodetector. (*Nat. Nanotechnol.* 2014, **9**, 814-819; *ACS nano* 2013, **8**, 216-221; *Nano Lett.* **2013**, 13, 358-363).

Comment 2. You mention e.g. night vision and sensing as potential applications. However, you use rather high laser powers and do not demonstrate the possibility to detect low intensity light contrary to those claims. Can you experimentally demonstrate the suitability of your detector for low light intensity applications?

Following the referee’s suggestion, we have experimentally demonstrated the suitability of our detector for low light intensity detection. (Page 9, Line 223-227)

We use the STO detector to detect the radiation from human body. When a human finger is placed above the suspended end with a separation of about 1.5 mm, the output voltage can be up to 0.93 mV. The sensitivity of STO photodetector on human radiation is much higher than graphene-based PTE detector, which only

generates the photovoltage on the order of nV (*Nano Lett.* **2015**, 15, 7211-7216).

Comment 3. You claim an ultra-broad spectral response from 325nm-10.67 μ m; however there is a large gap from 1550nm-10 μ m in your data. The claim is an overstatement without providing experimental evidence.

Due to the lack of relevant lasers, we did not provide the measurement data covering the spectral range from 1550 nm to 10 μ m. However, as can be seen from the absorption spectra, STO has three representative absorption bands, i.e., above-band gap absorption (200-400 nm), flat sub-band gap absorption (400 nm-8.7 μ m) and strong phonon absorption (8.7-12.3 μ m). By providing photodetection characterizations in these three regions, we think it is appropriate to refer to our STO detector as ultrabroadband photodetector.

Comment 4. What is your assumption to detect THz radiation based on? No references, etc given.

We have added the reference to support our opinion. (Page 3, Line 74-75)

Previous study shows that STO also has soft phonon mode at 2.62 THz, which indicates it is possible to use STO PTE detector to detect THz radiations.

Comment 5. You generally use graphene and 2D-materials as a comparison for the performance of your device. However, in line 75 you chose nanoporous Si as reference to underline the “high” speed of your device and omit graphene and 2D-materials. Instead of cherry picking, please also add the performance of graphene and 2D-materials to fairly compare your device performance.

Further comparison has been added (Page 9, Line 231-235, Supplementary Table 1).

The responsivity of STO photodetector is comparable to, and sometimes even better than those of 2D materials-based PTE detectors. On the other hand, STO photodetector can be used for detecting high-intensity light, while 2D materials-based

PTE detectors usually can only be used for low light intensity detections. However, because of the bulk nature of currently used STO photodetector, its response speed is much slower than those of 2D materials photodetectors. Thus, improving the response speed of STO detector will be our priority in the following work.

Comment 6. In line 95, you claim annealing leads to oxygen vacancies. However, neither experimental evidence nor a reference is provided.

A direct experimental evidence is that after annealing STO in reducing atmosphere, STO changes from insulator to semiconductor. The increased carrier concentration is related to the creation of oxygen vacancies. The corresponding defect equation written in Kroger–Vink notation is $O_O^{\times} = V_O^{\bullet\bullet} + 1/2O_2(g) + 2e'$, where O_O^{\times} represents oxygen ion sitting on an oxygen lattice site, $V_O^{\bullet\bullet}$ is an oxygen vacancy with +2 charge, and e' is an electron. As a result, the production of two free electrons is accompanied by one oxygen vacancy.

In the revised manuscript, we have added the relevant discussion and references (Page 4, Line 99-101).

Comment 7. Section photoresponse mechanism: You neither make a clear statement what the mechanism is nor do you present any convincing experimental data or theoretical modelling. The only statement is “a fast and slow component” and many guesses.

We have revised the section of photoresponse mechanism, and deleted the description about the fast and slow component. Now this section is divided into two parts. The first part is about the simultaneous measurements of temperature and photovoltage to prove the photothermoelectric mechanism. The second part is about the explanation of the hysteresis-like trace formed under laser heating (Figure 4c). We found this hysteresis-like trace is a common phenomenon for bulk materials. And we explain this phenomenon by considering that there exists difference between surface temperature and bulk temperature, which is a common phenomenon in photothermal effect. Details can be found in Page 7, Line 175-185.

Comment 8. Overall, the manuscript presents rather a collection and description of experimental data without clear explanation. It suffers from overstatements and lacks relevant comparisons and references. Further, the English language needs to be improved.

Following the referees' comments, we have made serious revisions and added more references. We hope that current version will reach the criterion of Nature Communications.

REVIEWERS' COMMENTS:

Reviewer #1 (Remarks to the Author):

The manuscript presents an interesting research work. This paper has been modified in general, but it is inaccurate and not active in describing other person's work, which we are not satisfied. There are still a couple of issues needed to be noted and it can be published after minor revisions: (1) The authors should positively describe the previous works finished by Others such as adding the sentences of (Ya Yang's research group has firstly developed the photo-induced thermo-phototronic effect and pyro-phototronic effect to realize the impressed light detection and largely enhance the performances of photodetectors, respectively.) The authors should revise the sentences in current manuscript in the introduction section and emphasize the importance of others' work.

Reviewer #2 (Remarks to the Author):

The authors have addressed my comments in their revised version of the manuscript. Therefore, I recommend its publication as it is.

Reviewer #3 (Remarks to the Author):

Dear authors,

My previous questions have been satisfactorily addressed and answered. The manuscript has improved significantly due to the additional data, revised text, added references and comparisons. Therefore, I do now agree to a publication in Nature Communications.

Reply to reviewers' comments: NCOMMS-18-08247A

Reviewer #1:

The manuscript presents an interesting research work. This paper has been modified in general, but it is inaccurate and not active in describing other person's work, which we are not satisfied. There are still a couple of issues needed to be noted and it can be published after minor revisions:

(1) The authors should positively describe the previous works finished by Others such as adding the sentences of (Ya Yang's research group has firstly developed the photo-induced thermo-phototronic effect and pyro-phototronic effect to realize the impressed light detection and largely enhance the performances of photodetectors, respectively.) The authors should revise the sentences in current manuscript in the introduction section and emphasize the importance of others' work.

We firstly appreciate the reviewer's positive comments on our work.

Based on the referee's advice, we have added the following sentence in the first paragraph of the introduction section. "Recently, the photo-induced thermo-phototronic effect⁵ and pyro-phototronic effect^{6,7} have been demonstrated to greatly enhance the performances of photodetectors."

Reviewer #2:

The authors have addressed my comments in their revised version of the

manuscript. Therefore, I recommend its publication as it is.

We appreciate the reviewer's positive comments on our work.

Reviewer #3:

My previous questions have been satisfactorily addressed and answered. The manuscript has improved significantly due to the additional data, revised text, added references and comparisons. Therefore, I do now agree to a publication in Nature Communications.

We appreciate the reviewer's positive comments on our work.